# Improved Prediction Accuracy for Late-Onset Preeclampsia Using cfRNA Profiles: A Comparative Study of Marker Selection Strategies

**DOI:** 10.3390/healthcare13101162

**Published:** 2025-05-16

**Authors:** Akiha Nakano, Kohei Uno, Yusuke Matsui

**Affiliations:** 1Biomedical and Health Informatics Unit, Nagoya University Graduate School of Medicine, Nagoya 461-8671, Japanuno.kohei.z8@f.mail.nagoya-u.ac.jp (K.U.); 2Institute for Glyco-Core Research (iGCORE), Nagoya University, Nagoya 461-8673, Japan

**Keywords:** late-onset preeclampsia, cell-free RNA, biomarkers, maternal immunity, machine learning, maternal factors

## Abstract

**Background**: Late-onset pre-eclampsia (LO-PE) remains difficult to predict because placental angiogenic markers perform poorly once maternal cardiometabolic factors dominate. **Methods**: We reanalyzed a publicly available cell-free RNA (cfRNA) cohort (12 EO-PE, 12 LO-PE, and 24 matched controls). After RNA-seq normalization, we derived LO-PE candidate genes using (i) differential expression and (ii) elastic-net feature selection. Predictive accuracy was assessed with nested Monte-Carlo cross-validation (10 × 70/30 outer splits; 5-fold inner grid-search for λ). **Results**: The best LO-PE elastic-net model achieved a mean ± SD AUROC of 0.88 ± 0.08 and F1 of 0.73 ± 0.17—substantially higher than an EO-derived baseline applied to the same samples (AUROC ≈ 0.69). Enrichment analysis highlighted immune-tolerance and metabolic pathways; three genes (HLA-G, IL17RB, and KLRC4) recurred across >50% of cross-validation repeats. **Conclusions**: Plasma cfRNA signatures can outperform existing EO-based screens for LO-PE and nominate biologically plausible markers of immune and metabolic dysregulation. Because the present dataset is small (n = 48) and underpowered for single-gene claims, external validation in larger, multicenter cohorts is essential before clinical translation.

## 1. Introduction

Preeclampsia is a condition characterized by the new onset of hypertension and proteinuria—or organ dysfunction such as liver or kidney impairment—after 20 weeks of gestation. It is reported to occur in approximately 2–5% of pregnancies worldwide [1]. This disorder significantly increases morbidity and mortality for both mothers and fetuses and can lead to preterm delivery or severe complications (e.g., HELLP syndrome).

Traditionally, preeclampsia has been categorized into early-onset (occurring before 34 weeks of gestation) and late-onset (occurring at or after 34 weeks of gestation) forms. Early-onset preeclampsia is typically associated with marked placental insufficiency and vascular dysfunction and tends to present with more severe clinical outcomes. In contrast, late-onset preeclampsia is thought to be more influenced by maternal factors (obesity, hypertension, metabolic risks, etc.) [2,3]. Although late-onset preeclampsia is often regarded as relatively mild, it still raises the risk of maternal–fetal complications and frequently necessitates cesarean delivery or other medical interventions.

Currently, the only definitive cure for preeclampsia is delivery, and effective pharmacological interventions remain limited—especially for late-onset cases. For instance, low-dose aspirin has been shown to significantly reduce the incidence of early-onset preeclampsia (before 34 weeks), but meta-analyses suggest that this prophylactic effect is less pronounced in late-onset disease [4]. Hence, early risk stratification and management of late-onset preeclampsia remain crucial.

Several screening approaches have been proposed to enable early risk assessment of late-onset preeclampsia, combining maternal background factors (e.g., chronic hypertension, obesity, and history of diabetes), uterine artery Doppler measurements, and serum biomarkers (e.g., PlGF and sFlt-1). However, changes in placental-derived factors are less pronounced in late-onset cases than in early-onset cases, and predictive models relying solely on placental angiogenic factors often reach a sensitivity of around only 40% [2,5] and, in systematic reviews, rarely exceed an AUROC of 0.70 for late-onset disease [3]. Consequently, late-onset preeclampsia has proven more challenging to predict with high accuracy. While it has been noted that maternal factors (e.g., high BMI and advanced maternal age) play a major role in late-onset cases [3,6], the specific molecular mechanisms underlying this subtype have not yet been fully elucidated.

Recent literature points to two converging biological axes in late-onset PE—failure of maternal–fetal immune tolerance and exacerbated third-trimester metabolic stress. Placental and review data highlight diminished HLA-G/KIR signaling and a Th17/IL-17-skewed cytokine milieu in LO-PE [7,8], while large cohort and cfRNA studies report the early enrichment of Allograft Rejection and Estrogen-response pathways [9]. Complementarily, metabolic-syndrome traits—insulin resistance, dyslipidemia, and altered glycolytic flux—have been proposed as principal maternal drivers of LO-PE [10]. Guided by this evidence, we focus our cfRNA feature-selection and downstream interpretation on genes mapping to immune-tolerance and metabolic pathways.

In recent years, machine learning (ML) and artificial intelligence (AI) approaches have gained attention for their potential to integrate these complex risk factors multidimensionally and are now highlighted as promising avenues for PE prediction in dedicated reviews [11]. For example, analyses of large-scale electronic health records (EHRs) incorporating diverse maternal background and laboratory data have demonstrated high-accuracy prediction with an AUC exceeding 0.9 [6]. Nonetheless, most studies to date are retrospective and confined to specific cohorts and thus lack external validation or prospective evaluation. Although attempts to merge multi-omics data (e.g., genetic risk scores, proteomics, and metabolomics) have been reported [12], the cost and clinical feasibility remain significant barriers.

Parallel to the surge in AI-driven risk models, cell-free RNA sequencing (cfRNA-seq) of maternal plasma has become a leading avenue for non-invasive biomarker discovery. Recent high-throughput studies using this technique repeatedly implicate two pathophysiologic axes in late-onset PE—(i) the loss of maternal–fetal immune tolerance, exemplified by downregulated placental HLA-G/KIR checkpoints and a Th17-skewed cytokine milieu [7,8], and (ii) third-trimester metabolic stress, marked by systemic insulin-resistance signatures and enrichment of glycolysis- and estrogen-response gene sets [9,10]. A 2024 systematic review synthesized these observations into a dual-hit model in which rising maternal metabolic load amplifies incipient immune dysfunction to trigger clinical LO-PE [3].

Circulating cfRNA originates from both maternal blood cells and the placenta; trophoblast-derived transcripts are detectable from the first trimester and increase steadily with gestation. Unlike cfDNA, cfRNA captures the moment-to-moment transcriptional state of maternal–fetal tissues, providing time-resolved insight into immune, angiogenic, and metabolic pathways [9]. Longitudinal, deep-coverage cfRNA-seq has shown that placental, endothelial, and leukocyte signatures begin to diverge weeks before symptom onset. Because LO-PE is driven more by maternal cardiometabolic stress and systemic inflammation than by early placental maldevelopment, cfRNA offers a unique window into these evolving maternal responses—signals that may be missed by placental protein biomarkers alone.

Early-Onset Preeclampsia (EO-PE): This form is predominantly driven by placental abnormalities and immune dysregulation that begin early in gestation; distinct differential expression of cfRNA has been reported. For example, Moufarrej et al. demonstrated a high-accuracy model (AUC ≈ 0.9) using cfRNA derived from maternal plasma, suggesting an impairment of immune response and angiogenic pathways [9].Late-Onset Preeclampsia (LO-PE): Maternal comorbidities such as obesity or chronic hypertension play a substantial role, often diminishing the utility of purely placental biomarkers for high-sensitivity prediction. Indeed, many studies investigating cfRNA- or metabolite-based tests focus on overall PE risk and do not provide separate metrics (e.g., AUC) for LO-PE alone. For example, while Maric et al. [13] report robust performance in predicting PE, their models do not isolate late-onset cases. As a result, the true accuracy for LO-PE remains unclear, and some data even suggest that maternal factors may overshadow direct placental signals, leading to potentially lower AUCs for late-onset compared to early-onset PE. Moving forward, it will be crucial to refine LO-PE-specific molecular signatures—possibly through multi-omics approaches integrated with maternal clinical data—and validate such signatures in large external cohorts. This line of research is expected to clarify whether dedicated LO-PE models can outperform current one-size-fits-all approaches and ultimately improve risk stratification in this patient population.

This study focuses on LO-PE and aims to (1) identify cfRNA-based biomarker candidates specific to LO-PE and (2) develop and evaluate machine learning models using these markers. More specifically, our objectives are:To characterize cfRNA profiles in LO-PE and compare them with known markers predominantly associated with EO-PE.To apply two feature selection strategies—(A) an approach based on differential expression analysis and (B) an approach leveraging prediction errors (via the elastic-net solution path)—and then assess LO-PE prediction performance in terms of AUC, sensitivity, and specificity.To examine the performance trade-offs involved in simultaneously predicting both EO- and LO-PE, and to investigate how immune tolerance and metabolic pathways might be affected.

Ultimately, this study seeks to elucidate the mechanisms underlying late-onset preeclampsia—particularly those related to immune modulation and placental invasion—by leveraging cfRNA signatures, with the goal of informing future clinical management of preeclampsia.

## 2. Materials and Methods

This study analyzed cfRNA sequencing data from a total of 48 samples, comprising EO-PE, LO-PE, and corresponding control groups for each subtype. Our goal was to identify potential biomarkers specifically associated with LO-PE and then construct and evaluate a diagnostic prediction model. The overall analytical workflow is illustrated in Figure 1.

### 2.1. Dataset

The dataset is based on a cfRNA cohort described in Reference [14], which includes 12 subjects with LO-PE, 12 subjects with EO-PE, and 12 controls for each group. All participants carried singleton pregnancies without structural fetal anomalies and were recruited prospectively at Stanford University Medical Center between 2014 and 2017 under IRB protocol #28979. Maternal blood was drawn at the time of clinical diagnosis (mean ± SD gestational age: EO-PE 29.2 ± 2.3 weeks; matched EO-controls 29.3 ± 2.3 weeks; LO-PE 35.6 ± 1.3 weeks; matched LO-controls 35.9 ± 0.8 weeks). The 24 normotensive controls were selected from a single pool and randomly assigned 1:1 to the EO and LO subgroups so that gestational age matching with their respective case groups was preserved. Written informed consent for cfRNA sequencing and data deposition in dbGaP (accession phs002017.v1) was obtained from every participant in accordance with the Declaration of Helsinki. Blood samples were collected at the time of PE diagnosis, and cfRNA (cell-free RNA) was extracted from maternal plasma for Next-Generation Sequencing (NGS). Because the resulting RNA reads may include transcripts of both placental and maternal origin, it offers the intriguing possibility of capturing both maternal and placental factors. Given that our analysis involves a relatively small sample of 48 total specimens, special attention must be paid to sample-size limitations, the risk of overfitting, and the need for further external validation when constructing prediction models. Because the published study supplies an already aligned gene-count matrix, our workflow starts from this matrix; additional read-level QC or alignment was not required. Participant demographics are summarized in Appendix A.

### 2.2. Strategy for Selecting Signature Genes

To identify biomarkers specific to LO-PE, we employed two approaches: differential expression gene (DEG) and an elastic-net-based machine learning method leveraging prediction error. Differential expression (DE) analysis tests whether the mean read count for each gene differs between conditions. First, we used RNA-seq count data to conduct three intergroup comparisons: “early-onset vs. control”, “late-onset vs. control”, and “early-onset vs. late-onset”. We then used edgeR [15,16] and limma [17,18] to extract genes showing statistically significant differential expression. This process involved adjusting the *p*-values via the Benjamini–Hochberg method [19] to control the false discovery rate (FDR) and using log2 fold change (logFC) values as an additional criterion for candidate gene selection. By excluding genes that were differentially expressed in both early- and late-onset groups, we obtained a set of candidate genes more specific to LO-PE.

Next, we applied an elastic-net regression model using the glmnet [20] package, which implements coordinate-descent optimization [20], to tackle two classification tasks—“control vs. early-onset” and “control vs. late-onset”. We optimized the model’s hyperparameter, λ, through cross-validation to maximize prediction performance (AUC) and simultaneously minimize the number of genes used. By examining the solution path, we extracted genes that contributed most significantly to predicting LO-PE and designated them as late-onset-specific signatures for subsequent functional analysis. Since the elastic net combines both $L_{1}$ (Lasso) and $L_{2}$ (Ridge) regularization, it effectively prevents overfitting and performs variable selection automatically. This makes it particularly useful for scenarios, such as ours, where one must narrow down important features from a large pool of genes.

### 2.3. Building and Evaluating the Predictive Model

Using the selected signatures, we constructed models to predict LO-PE and evaluated their classification accuracy. Model performance was assessed with a nested Monte-Carlo cross-validation (MC-CV) procedure. A stratified 70/30 split was repeated 10 times (outer loop); within each outer training set, a 5-fold inner CV tuned the elastic-net λ across a 50-value grid. We trained an elastic-net model separately on the “early-onset signature” and the “late-onset signature” and then computed the AUC to assess performance for the classifications “control vs. late-onset” and “control vs. early-onset”. In addition, we evaluated its performance when combining the early-onset and late-onset signatures to investigate whether handling both simultaneously would induce any performance trade-off. The AUC (Area Under the ROC Curve) serves as a comprehensive measure of a model’s ability to discriminate between true positives and false positives, with 1.0 indicating perfect accuracy and 0.5 indicating performance equivalent to random guessing. Where necessary, we also considered sensitivity and specificity to gain insight into the balance between false positives and false negatives.

### 2.4. Searching for Biomarker Candidates

We further investigated the late-onset signature genes extracted via prediction-error analysis by conducting gene set and pathway analyses to clarify their functional characteristics. Specifically, we cross-referenced the gene lists with databases such as Gene Ontology and KEGG [21] to statistically evaluate the enrichment of pathways related to metabolism, immunity, and other processes, using Fisher’s exact test. Of particular interest were genes involved in immune tolerance or placental invasion, such as HLA-G and IL17RB; their inclusion in the signature could suggest associations with maternal immune dysregulation or trophoblast (EVT) dysfunction in LO-PE. We compared such findings against previous studies to explore their biological significance. Ultimately, this functional validation of late-onset-specific gene groups helps lay the groundwork for determining their potential clinical utility as diagnostic biomarkers in future research.

### 2.5. Performance Metrics

Classifier performance was evaluated with the following standard measures:(1)$$\mathrm{AUROC}=\int_{0}^{1}TPR\left(\alpha \right)\;d\alpha$$(2)$$\mathrm{Sensitivity}\;(\mathrm{Recall})=\frac{TP}{TP+FN},\;\;\mathrm{Specificity}=\frac{TN}{TN+FP}$$(3)$$\mathrm{Precision}=\frac{TP}{TP+FP},\;\;F_{1}=\frac{2\;\mathrm{Precision}\times \mathrm{Sensitivity}}{\mathrm{Precision}+\mathrm{Sensitivity}}$$
where TP, TN, FP, and FN denote true positives, true negatives, false positives, and false negatives, respectively. AUROC provides an overall measure of discrimination; the $F_{1}$-score complements AUROC by balancing Precision and Sensitivity, which is useful when class sizes are imbalanced.

### 2.6. Model Evaluation

Finally, to benchmark the proposed elastic net against alternative learners, we repeated the entire nested Monte-Carlo pipeline with Random Forest, linear SVM, and XGBoost (hyper-parameters tuned in the inner loop). Mean ± SD AUROC and F1 are reported in Appendix A, and pooled ROC curves are shown in Appendix A.

## 3. Results

### 3.1. Identification of Signature Genes and Feature Selection

Appendix A summarizes maternal age, body mass index (BMI), and gestational age at sampling for each study group. Groups did not differ in age or gestational week (all *p* > 0.5), whereas BMI was significantly higher in the LO-PE group compared with its matched controls (*p* = 0.036) (Appendix A). First, we performed differential expression analyses (DEG) for three comparisons—(1) early-onset PE vs. control, (2) late-onset PE vs. control, and (3) early-onset PE vs. late-onset PE—and generated lists of signature candidates by systematically varying the thresholds for *p*-values and log fold change (logFC) (Figure 1, panel 1a-1). We tested three cutoff conditions: (A) *p* < 0.05, (B) *p* < 0.05 and |logFC| > 1, and (C) *p* < 0.01 (1a-1 in Figure 1). As shown in Figure 2A, the late-onset-specific signatures comprised 64 genes under condition A, 1 gene under condition B, and 7 genes under condition C. In contrast, the early-onset-specific signatures included 1334 genes (A), 11 genes (B), and 295 genes (C).

When we used these results to train an elastic-net model, the model that relied solely on the single gene KLRC4 (from condition B) attained an extremely high predictive performance for LO-PE (*AUC* = 1.0). A logistic model based on the single gene KLRC4 yielded an apparent AUROC = 0.875. A 1000-iteration label-permutation test produced a null AUROC distribution centered at 0.874 ± 0.004; the observed AUROC of 0.875 lay well within this range (one-sided *p* = 0.997). The full distribution is visualized in Appendix A, underscoring that the single-gene signal is indistinguishable from chance.

Benchmarking against three conventional classifiers confirmed the superiority of the elastic net: for EO-PE the elastic net reached AUROC 0.91 vs. 0.87 (RF), 0.83 (SVM), and 0.74 (XGBoost); for LO-PE, the corresponding figures were 0.83, 0.71, 0.73, and 0.61, respectively (Appendix A).

### 3.2. Candidate Selection via Prediction Error (Elastic-Net Solution Path)

Next, we performed cross-validation on the elastic-net model while varying the hyperparameter *λ* in 50 increments, adopting the parameter setting that maximized predictive performance (AUC) while minimizing the number of selected genes (1b in Figure 1). This approach extracted 52 genes as late-onset-specific signatures and 5 genes as early-onset-specific signatures (Figure 2B–D). These sets exhibited very little overlap with the signatures identified via DEG analysis. As shown in the Venn diagram in Figure 3, most genes from the solution path (SP)-based signatures and those from the DEG-based approach did not overlap for both early- and late-onset PE, indicating that the two methods complement each other.

### 3.3. Comparative Performance of Prediction Models

Table 1 and Figure 4 (ROC curves) present the results of elastic-net regression models constructed and validated using each signature set (DEG-based: A, B, C; elastic-net-based: SP). As a baseline, we also examined models without any signature genes:Training on early-onset samples yielded an AUC of 0.9375 for predicting early-onset PE but only 0.6875 for predicting late-onset PE.Training on late-onset samples resulted in an AUC of 0.6875 for predicting late-onset PE.

These findings indicate that the baseline approach is insufficient for predicting late-onset PE with high accuracy. In contrast, introducing late-onset-specific signatures (e.g., the 64 genes from condition A or the single gene from condition B) increased the AUC for late-onset PE prediction to around 0.88–1.0, highlighting the importance of markers specific to the late-onset subtype. Notably, the model using only KLRC4 (condition B) achieved an AUC of 1.0, but its generalizability remains uncertain. Additionally, when early-onset and late-onset signatures were used together, late-onset AUC sometimes decreased, and a drop in early-onset performance was also observed—indicative of a trade-off.

As shown in Table 1, the major differences in pathophysiology and molecular mechanisms between early- and late-onset PE make it challenging for a single signature to achieve high accuracy for both subtypes. Indeed, while adding late-onset-specific signatures improved the AUC for late-onset PE, it sometimes slightly reduced performance for early-onset predictions. These findings underscore previously reported observations that without tailored markers for late-onset PE, it is difficult to achieve high prediction accuracy.

### 3.4. Candidate Biomarkers and Functional Analysis

Based on an over-representation analysis of the 87 late-onset-specific genes (Figure 5), the most significantly enriched themes were (i) pro-inflammatory and innate-immune signaling—notably interferon-γ/α and TNF-α → NF-κB cascades, together with adaptive-immune terms such as allograft-rejection/graft-vs.-host pathways—and (ii) extracellular-matrix remodeling and cell-metabolic processes, including elastic-fiber assembly and heme metabolism.

Notably, HLA-G, IL17RB, and KLRC4—genes previously implicated in immune tolerance and trophoblast invasion—showed marked expression differences in the LO-PE group compared with controls. This observation suggests a potential role for impaired maternal–placental interactions.

## 4. Discussion

This study is constrained by its modest cohort size (12 LO-PE, 12 EO-PE, and 24 controls). An a priori power calculation shows that for 12 vs. 12 samples and a Cohen’s *d* = 0.8, a two-sided α = 0.05 yields power ≈ 0.47 (β ≈ 0.53), confirming that the dataset is under-powered for stable single-gene inference. Consistent with this, a 1000-iteration permutation test indicated that the KLRC4 single-gene model does not outperform chance (*p* = 0.997; Appendix A), underscoring the risk of over-fitting. No public cfRNA dataset with late-onset PE labels is currently available, so external replication remains an essential future step.

When the early-onset cfRNA signature was naïvely applied to the 12 LO-PE samples, discrimination was modest (AUROC ≈ 0.69); this serves as our internal baseline. Incorporating a late-onset-specific signature raised performance to AUROC = 0.88–1.00 under nested MC-CV, demonstrating a clear gain over the EO-baseline despite the small cohort.

For clinical context, protein/Doppler screens reach lower or comparable accuracy: in a first-trimester cohort, Tan et al. [5] reported AUROC = 0.744 (0.776 with MAP), while a third-trimester screen by Andrietti et al. [22] achieved AUROC = 0.881 (0.902 with MAP). These figures derive from different cohorts, time-points, and analytes (PlGF ± UtA-PI ± MAP) and therefore do not constitute a head-to-head comparison, but they indicate the present clinical performance ceiling. Genomics-assisted ML models that blend polygenic risk scores with routine clinical factors reach a similar range (AUROC ≈ 0.83) [12]. Taken together, these benchmarks suggest that cfRNA signatures—once validated in larger cohorts—could provide a non-invasive alternative with competitive, and potentially earlier, discrimination for late-onset PE.

Among the immune- and hormone/metabolic pathways, HLA-G and IL17RB appear particularly relevant in LO-PE. Wedenoja et al. [8] showed that HLA-G is significantly downregulated in preeclamptic placentas, indicating impaired fetal immune tolerance and reduced EVT infiltration—both hallmarks of shallow placental invasion. Likewise, IL17RB (the IL-25 receptor) fosters trophoblast proliferation; Liu et al. [23] reported that diminished IL-17RB expression in PE placentas correlates with suboptimal placental development. Our findings reinforce that a late-onset immune “collapse” may be tied to early disruptions in maternal–fetal tolerance. Moreover, Ma et al. [24] identified maternal KIR2DL4–fetal HLA-G genotype combinations that modulate preeclampsia risk, underscoring the genetic dimension of immune tolerance. Altogether, these data underscore how dysregulated HLA-G, IL17RB, and related genes (e.g., KLRC4) may drive LO-PE pathophysiology.

Previous placental-tissue studies have linked reduced HLA-G expression to inadequate EVT invasion in early-onset PE, but evidence in late-onset disease is scant and restricted to small immunohistochemistry series [8]. Our plasma-cfRNA data extend these observations by showing a systemic downregulation of HLA-G transcripts in LO-PE, detectable at the time of diagnosis. Similarly, IL17RB has been reported as a trophoblast-proliferation receptor in experimental work [23], yet has not, to our knowledge, been quantified in circulating RNA from LO-PE pregnancies. Finally, KLRC4 (NKG2F) appears only once in the PE literature (as a placental mRNA outlier in EO-PE); its repetition across five of the ten Monte-Carlo signature extractions suggests it may serve as a novel peripheral marker of late-stage maternal immune activation. Together, these results support the recently proposed ‘dual-hit’ model—immune tolerance collapse compounded by metabolic stress—in LO-PE [10], and they illustrate how cfRNA can surface candidate genes that traditional biomarker panels overlook.

Although various immune and metabolic pathways have been proposed in late-onset preeclampsia (LO-PE), direct evidence for specific processes—such as Allograft Rejection, Estrogen Response, or Glycolysis—and for genes like HLA-G, IL17RB, and KLRC4 remains limited. Recent cfRNA-based studies nonetheless suggest that maternal–placental signaling abnormalities can be detected earlier than clinical onset, potentially offering a broader diagnostic window for LO-PE risk [9,13]. However, the current data primarily indicate general immune dysregulation rather than a uniquely “late-onset-specific” mechanism. Further validation—for example, in multi-ethnic cohorts and via single-cell or multiomics approaches—will be crucial to pinpoint precisely which genes or pathways diverge in LO-PE.

This study observed a trade-off wherein including both early- and late-onset subtypes in a single model caused decreased accuracy for at least one subtype. That outcome likely stems from the distinct etiological underpinnings of EO-PE vs. LO-PE [25]. As noted by Moufarrej et al. [9], while EO-PE exhibits prominent signals from the placental formation stage, LO-PE often entails maternal metabolic and immune dysfunction that becomes clinically apparent later in gestation. Accordingly, future research should focus on (1) dynamic risk models incorporating longitudinal data by gestational week or (2) algorithms that screen early- and late-onset cases separately and then generate an integrated risk score.

The present study is subject to several limitations and suggests directions for future research. Consequently, prospective cohorts will be required to verify whether circulating HLA-G, IL17RB, and KLRC4 decline before symptom onset, thereby establishing them as early-warning markers rather than late epiphenomenal markers. First, each group comprised only 12 samples, emphasizing the need for validation in larger cohorts and multi-center collaborations to ensure the generalizability of these findings. Second, the predictive signatures identified in this study are sensitive to the choice of thresholds in differential expression analyses (*p*-values) and the setting of hyperparameters (λ in the elastic-net model). Comparative evaluations under multiple conditions are therefore necessary to confirm the robustness of the proposed signatures.

Finally, there are challenges to clinical implementation, as measuring cfRNA and conducting NGS analyses remain expensive and require specialized equipment. Standardizing sample processing and streamlining analytic protocols will be critical for broader clinical adoption. Moreover, integrating other omics data (e.g., metabolomics and epigenomics) to provide a more comprehensive assessment of both maternal and placental status is an important next step.

## 5. Conclusions

This study has three principal limitations: (i) the cohort is small (n = 48), limiting statistical power; (ii) model selection on a modest dataset raises a non-trivial risk of over-fitting; and (iii) no external cfRNA dataset with late-onset PE labels is yet available for validation. These caveats temper generalizability but also define clear priorities for future work. This study identified late-onset-specific cfRNA signatures and demonstrated that incorporating them into an elastic-net model substantially boosts predictive performance. Abnormalities in immune tolerance and metabolic systems—beyond what conventional early-onset markers can detect—may underlie the pathology of LO-PE. At the same time, challenges remain regarding sample size and model generalizability, pointing to the need for large-scale longitudinal studies and multi-omics integration. Ultimately, leveraging cfRNA-seq-based composite maternal–placental biomarkers in tandem with AI could significantly advance the early diagnosis and management of LO-PE.

## Figures and Tables

**Figure 1 healthcare-13-01162-f001:**
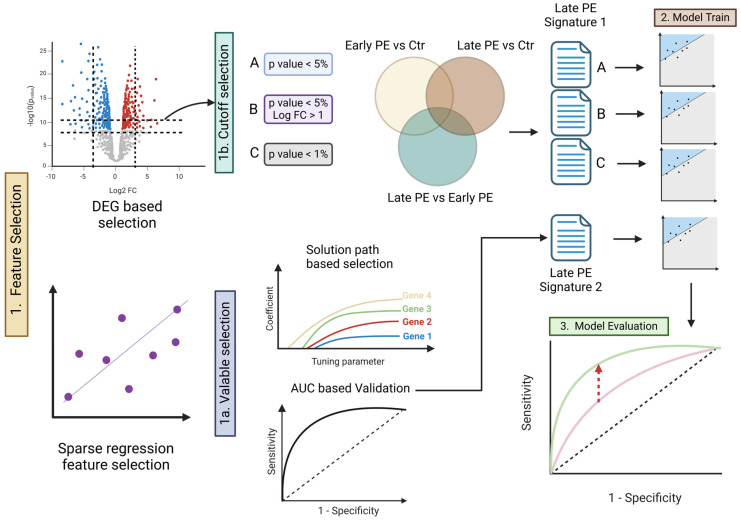
Schematic of the analytical workflow. Panel 1 (feature selection): cfRNA features were prioritized by two complementary routes. 1a Sparse-regression (elastic net) solution paths identified genes whose coefficients remained non-zero across a 50-value λ grid, and the resulting models were validated by AUC. 1b Differential expression (DEG) analysis compared Early-PE vs. Control, Late-PE vs. Control, and Early- vs. Late-PE; three cut-off schemes were explored—A *p* < 0.05, B *p* < 0.05 and |log FC| > 1, and C *p* < 0.01—yielding signature candidates shown in the Venn diagram. In the volcano plot, red dots indicate significantly up-regulated genes, blue dots down-regulated genes, and grey dots non-significant genes. Panel 2 (model training): for each DEG-derived signature (A–C) and for the elastic-net signature, separate elastic-net classifiers were trained on the Late-PE training data. Panel 3 (model evaluation): nested Monte-Carlo cross-validation supplied independent test sets; ROC curves illustrate that Late-PE-specific signatures (green) outperform naïvely transferred Early-PE signatures (pink), while the diagonal denotes chance.

**Figure 2 healthcare-13-01162-f002:**
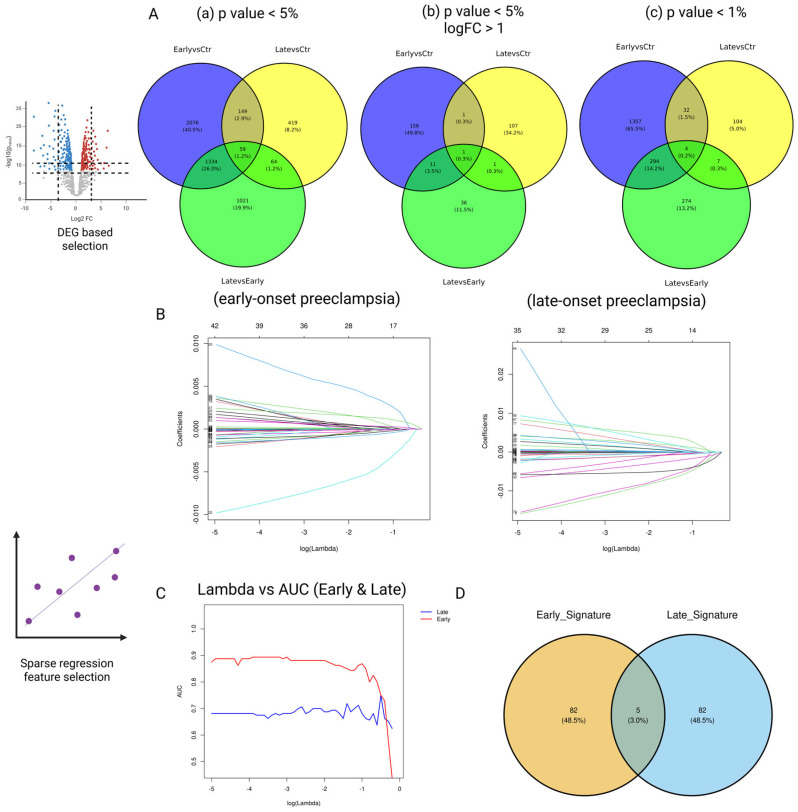
Multi-stage feature selection and signature derivation for early- (EO-PE) and late-onset pre-eclampsia (LO-PE). (**A**) Differential expression (DEG) stage. Venn diagrams show EO- and LO-specific gene sets obtained under three progressively stringent cut-offs: (**a**) *p* < 0.05; (**b**) *p* < 0.05 and |log_2_FC| > 1; (**c**) *p* < 0.01. Numbers indicate genes unique to, or shared between, the three pairwise contrasts (Early vs. Ctrl, Late vs. Ctrl, Late vs. Early). (**B**) Elastic-net solution paths. For each subtype, the coefficients of all candidate genes are traced across the log_10_(λ) grid; colored curves enter the model as λ decreases, illustrating automatic sparsification. (**C**) Model-selection curve. Mean outer-loop AUROC vs. log_10_(λ) for EO- (blue) and LO-PE (red). Plateaus mark the λ values giving maximal discrimination with minimal complexity. (**D**) Final signatures. Intersecting the optimal EO and LO coefficient sets yields five shared genes (center), with 82 genes retained exclusively in each subtype-specific signature.

**Figure 3 healthcare-13-01162-f003:**
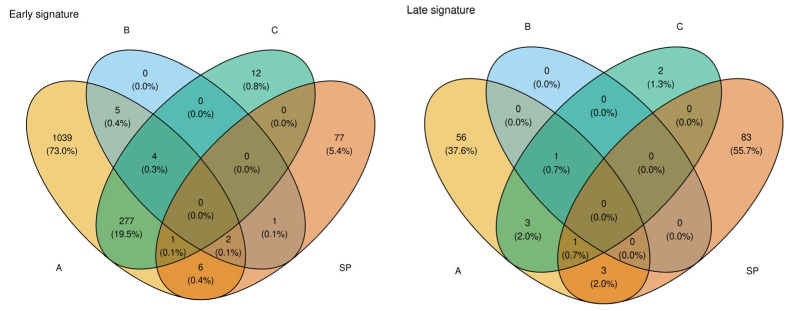
Overlap of gene signatures obtained with four selection strategies for (**left**) early-onset and (**right**) late-onset pre-eclampsia. Each Venn diagram contrasts the differential expression cut-offs A (*p* < 0.05), B (*p* < 0.05 and |log_2_FC| > 1), and C (*p* < 0.01) with the sparsity-optimized elastic-net solution path (SP). Numbers denote the absolute gene count and, in parentheses, the percentage of the total signature size for that subtype. The minimal intersection—five shared genes in the early set and three in the late set—underscores that the early- and late-onset signatures are largely distinct, supporting divergent molecular mechanisms between the two PE subtypes.

**Figure 4 healthcare-13-01162-f004:**
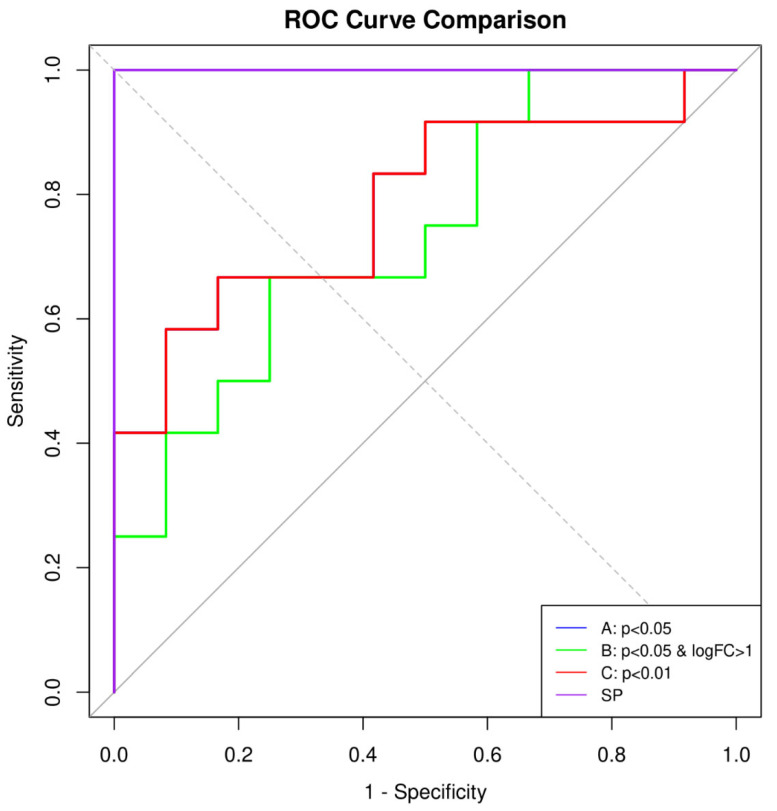
ROC curves from 10 repeats of Monte-Carlo cross-validation comparing three late-onset PE gene-signature models. The green, red, and violet lines correspond to signatures derived from cut-off B (*p* < 0.05 and |log_2_FC| > 1), C (*p* < 0.01; identical to A and therefore shown only once), and the sparsity-optimized elastic-net solution path (SP), respectively. Curves are obtained by aggregating predictions from the outer 70/30 test folds (n = 10). The elastic-net model (violet) maintains the highest sensitivity across the entire false-positive-rate range, confirming its superior AUROC in Table 1. Abbreviations: FPR, false-positive rate; TPR, true-positive rate.

**Figure 5 healthcare-13-01162-f005:**
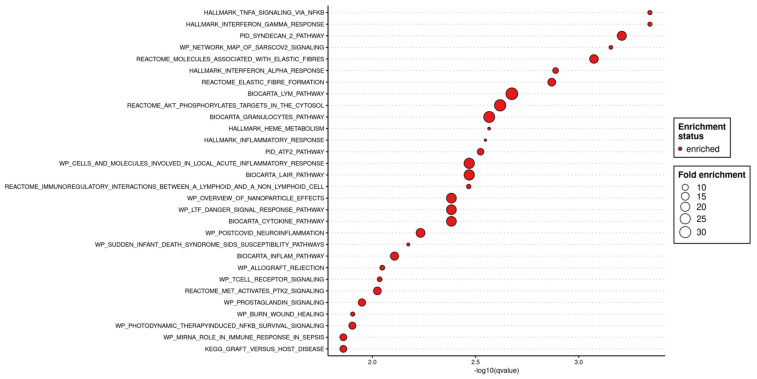
Results of pathway enrichment analysis for the 87 late-onset signature genes identified by the elastic-net model. The horizontal axis represents −log10(*p*-value), while bubble size corresponds to fold enrichment. Estrogen response, rejection-response pathways, and glycolysis/ketone-body metabolism pathways appear among the top enriched categories.

**Table 1 healthcare-13-01162-t001:** Discriminatory performance of gene-signature models for early- (EO-PE) and late-onset pre-eclampsia (LO-PE). Models were trained on either the EO or LO subset (columns 2–3) and evaluated on both subtypes (columns 4–5). “Prediction without signature” uses all expressed genes; the three DEG signatures correspond to cut-offs (A) *p* < 0.05, (B) *p* < 0.05 and |log_2_FC| > 1, and (C) *p* < 0.01; “Elastic-net signatures” are the sparse sets selected by nested Monte-Carlo cross-validation.

Prediction Without Signature		Training Data	Early	Late	Early	Late
		Early prediction AUC	0.9375	0.6875		
		Late prediction AUC	0.6875	0.6875		
		**Signature**	**Early**	**Late**	**Early and Late**	
**DEG Signature**						
(a) *p* value < 0.05						
Number of Early Signature	1334	Early prediction AUC	0.9944	0.6826	0.9938	0.6493
Number of Late Signature	64	Late prediction AUC	0.6625	0.9563	0.65	0.6979
(b) *p* value < 0.05 and logFC > 1						
Number of Early Signature	11	Early prediction AUC	0.915	0.653	0.905	0.738
Number of Late Signature	1	Late prediction AUC	0.687	0.736	0.68	0.786
(c) *p* value < 0.01						
Number of Early Signature	295	Early prediction AUC	0.997	0.66	0.995	0.606
Number of Late Signature	7	Late prediction AUC	0.67	0.755	0.656	0.651
**Elastic Net–Based Signatures**						
Number of Early Signature	87	Early prediction AUC	0.988	0.828	0.869	0.856
Number of Late Signature	87	Late prediction AUC	0.71	0.988	0.687	0.869

## Data Availability

The sequencing data reanalyzed in this study are publicly available at NCBI dbGaP (accession number phs002017.v1.p1) and in the Appendix A of [https://doi.org/10.1126/scitranslmed.aaz0131]. All newly derived data (including gene signature lists and statistical outputs) are presented in the article. Further inquiries can be directed to the corresponding author.

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
