# Peer review of "Improved Prediction Accuracy for Late-Onset Preeclampsia Using cfRNA Profiles: A Comparative Study of Marker Selection Strategies"

_healthcare, 2025, doi:10.3390/healthcare13101162_

Round 1
Reviewer 1 Report
Comments and Suggestions for Authors
This is a promising and well-designed study, but it requires clearer methodological detailing and a more grounded discussion of its clinical implications and limitations. The strength of the conclusions depends heavily on acknowledging the constraints posed by the small sample size and lack of external validation.
- An AUC of 1.0 is reported using a single marker (KLRC4) without external validation.
Add a more detailed cautionary note regarding the high risk of overfitting, particularly due to the small sample size (n=12 LO-PE cases). A comparison with similar studies should also be included to contextualize the performance.
- Only 48 samples were analyzed (12 per group), which limits statistical power.
This limitation should be emphasized earlier in the Discussion section, not only at the end. The need for replication in larger cohorts should be clearly and strongly argued.
- The manuscript does not provide demographic or clinical characteristics of the study participants (e.g., age, BMI, ethnicity). Include a descriptive table summarizing the clinical background of participants, even if based on data from the original referenced study, as such variables are known to influence LO-PE risk.
- While the manuscript mentions that current models achieve an AUC of ~0.69, no direct comparison is presented (e.g., with PlGF + Doppler models).
Add a comparative overview (table or narrative) highlighting the performance of the proposed cfRNA model versus current clinical standards. - The procedure for splitting the data into training and test sets is not clearly described (e.g., random? stratified?). Clarify the randomization strategy, and indicate whether full cross-validation or repeated holdout validation was performed beyond λ tuning.
- Figure 1 is briefly mentioned but not well-integrated into the narrative. Provide a clear reference to each panel (1a, 1b, etc.) within the main text, and explain their significance in context.
Reviewer 2 Report
Comments and Suggestions for Authors The study has leveraged cell-free RNA (cfRNA) sequencing of maternal plasma in 48 samples—comprising early-onset PE, late-onset PE, and corresponding control groups—to identify LO-PE–specific biomarkers.Differential expression analyses and elastic net regression have been used to extract LO-PE gene signatures, with solution paths guiding the selection of the most predictive features. A comparative study of marker 3 selection strategies have been presented.
To improve the quality of the manuscript, my recommendations are given as below;
-Formulas and explanations of performance metrics must be presented.
-Methods must be formulated or illustrated with a flow chart or figure.
-In addition to cell-free RNA (cfRNA) sequencing, several alternative and complementary methods can be employed to improve the prediction and understanding of late-onset preeclampsia (LO-PE). These include both experimental and computational strategies that offer a more holistic view of the disease's molecular and clinical landscape:
1. Multi-omics Approaches
2. Machine Learning and Artificial Intelligence
3. Integration of Clinical Data
4. Exosomal RNA Profiling
5. Single-cell RNA Sequencing (scRNA-seq)
6. Longitudinal (Time-Series) Analysis
7. Network Analysis and Causal Inference
The subject of Machine Learning and Artificial Intelligence is the most popular approach in this list.
You must present Machine Learning and Artificial Intelligence methods and their comparisons for your experimental studies. For example; you can use Random Forest, XGBoost, SVM, Ensemble Learning strategies or Deep Learning (e.g., RNNs, Transformer-based models)
-References must be up-to-date and the number of articles should be increased.
The English could be improved to more clearly express the research.
Reviewer 3 Report
Comments and Suggestions for Authors
In the manuscript, Dr. Nakano and other co-authors leveraged cell-free RNA (cfRNA) sequencing of maternal plasma to compare early-onset PE, late-onset PE and corresponding control group, aiming to identify late-onset PE specific biomarkers. Authors utilized various advanced approaches such as differential expression analysis, machine-learning models and pathway analysis to identify biomarkers and access prediction performance. In general, the article is of great interest, innovation and scientific soundness to readers. However, this manuscript is not well-presented and lacks necessary citations, introduction to methods and supplemental materials. I have multiple major concerns as follows:
- In Introduction and Methods parts, the authors need to provide citations to some important methods and introductions, which include:
- Line 45: the risk of maternal-fetal complications
- Lines 91-104: Late-Onset Preeclampsia
- Line 159 Lasso NAD Ridge regression
- Line 182-192 pathway enrichment analysis
The authors should note that not all audience of this Journal have a well understanding of bioinformatics and genomic researches. Thus, it’s important to go through your manuscript and include all necessary citations of all advanced methods used in the study.
- In Introduction and Methods parts, the authors need to provide more details about specific methods and techniques used in the study, which include:
- Lines 76-83: detailed introduction of cfRNA and its application, particularly in maternity and preeclampsia.
- Since authors used differential expression analysis and elastic net regression, I would suggest authors including a few sentences to introduce them in either Introduction or Method part, i.e. what’s differential expression analysis, how elastic net regression differs from other traditional modeling approaches like logistic regression, etc. Please do include citations.
- In Methods part line 131, instead of mentioning that cfRNA cohort is described in Reference, the authors need to have a few sentences introducing this cohort and why you used 48 subjects from this cohort.
- Any discussion about the limitations of this study, such as small sample size, should only be put in the Conclusion part.
- Just a question, in selecting signature genes and building the predictive model, the authors used different validation methods: cross-validation and holdout method. I would like to ask the authors why used different validation methods for the above two steps.
- Results are not well organized. I acknowledge that authors conducted lots of analyses for each step you mentioned in Methods. However, given the limited space, it is hard to put everything in the main text and will create confusion for audiences to understand and get useful information with large amounts of emerged results. Thus, I have the following suggestions:
- Attach the supplementary materials and include any tables and figures which supplement main results, such as cross-validation results about lambda, ROC curves (e.g. multiple ROC curves in one plot with AUC values, Table 1 is not straightforward for comparison), etc.
- Figure 1 is not clear to me. The authors should explicitly provide alphabets and numbers for each individual plot. More explanation for each individual plot should be added for Figure 1.
- Figure 3 is not very clear to me. Roc_A, B, and C are fully overlapping with the line of AUC=0.5. Does this suggest that the three approaches based on DEG did not have any prediction ability and are the same as random guessing? I would suggest adding more ROC curves with AUC > 0.5 in this plot for comparison purposes, and rest of them could be given in supplemental.
- A typo in line 212: should be Figure 2B-D, not Figure 3B-D.
I really like this manuscript in terms of its finding and scientific soundness. However, the quality of this manuscript must be improved and more details should be included. Please go through the manuscript and revise as much as possible. Improving the quality of the exposition can make an interesting study appealing to more readers.
Reviewer 4 Report
Comments and Suggestions for Authors
The manuscript attempted to touch very important and interesting topic. Identification of new reliable molecular markers for early prediction of preeclampsia is highly required. Although the search for preeclampsia predictors is an extensive field of research, the authors tried to shed light on the problem from a new angle focusing on late onset preeclampsia. The work appears to be well done and written. Importantly, authors clearly defined the limitations of their work. However, I have a few minor comments which should be addressed before considering the paper for publication:
- Introduction, lines 115-122: It is not clear why immune tolerance and metabolic pathways have been chosen for analysis. Please, specify.
- Materials and Methods, lines 131-132: It is stated that 12 subjects with LO-PE, 12 subjects with EO-PE, and 12 controls for each group were included in analysis. Does it mean the control individuals were different for EO-PE and LO-PE groups? If so, what was the difference? Gestational age at the time of blood withdraw? Time of delivery? Please, provide detailed information. The same concerns the rationale of using different participants and criteria of selection.
- Discussion: please, indicate whether the immune and metabolic pathways, Allograft Rejection Estrogen Response, Glycolysis, as well as HLA-G, IL17RB and KLRC4 genes have been proposed as the factors implicated in LO-PE in earlier studies, or this is the first finding.
Round 2
Reviewer 2 Report
Comments and Suggestions for Authors
I appreciate the authors’ comprehensive response to my comments and commend the substantial improvements made in the revised manuscript. All requested changes have been appropriately addressed:
-
The performance metrics are now clearly defined with corresponding formulas and explanations.
-
The methodology has been effectively clarified through an improved schematic figure.
-
The inclusion of benchmarking against alternative machine learning models (Random Forest, SVM, XGBoost) provides valuable context and supports the robustness of the proposed approach.
-
Recent and relevant literature has been added to strengthen the background and discussion sections.
-
The manuscript has been formatted in full compliance with journal guidelines, and the language has been polished for clarity and fluency.
Overall, the revised version demonstrates increased scientific rigor and clarity. I find the manuscript suitable for publication in its current form.
Reviewer 3 Report
Comments and Suggestions for Authors
The authors has successfully addressed all of my major comments. Now the quality of the manuscript has been significantly improved, and I think the manuscript now meet the requirement of the journal and I recommend accept in the current version.